# Manipulated Oral and Rectal Drugs in a Paediatric Swedish University Hospital, a Registry-Based Study Comparing Two Study-Years, Ten Years Apart

**DOI:** 10.3390/ph16010008

**Published:** 2022-12-21

**Authors:** Åsa C. Andersson, Staffan Eksborg, Ulrika Förberg, Per Nydert, Synnöve Lindemalm

**Affiliations:** 1Karolinska University Hospital, 17164 Stockholm, Sweden; 2Department of Women’s and Children’s Health, Karolinska Institutet, 17177 Stockholm, Sweden; 3School of Health and Welfare, Dalarna University, 79131 Falun, Sweden; 4Department of Clinical Science, Intervention and Technology, Karolinska Institutet, 14152 Stockholm, Sweden

**Keywords:** manipulation, paediatric, medicine, dosing accuracy, dosage form, oral, rectal

## Abstract

This is a registry-based study with the aim of describing and comparing the frequency of manipulations of solid oral and rectal medicines in 2009 and 2019 at inpatient units and an emergency department in a paediatric hospital within a Swedish university hospital. All patients aged 1 month–18 years with oral or rectal administrations were included. In total, 140,791 oral and rectal administrations were included in 2009, and 167,945 oral and rectal administrations were included in 2019. The frequency of patients receiving at least one manipulated oral medicine decreased between the study years, both in inpatient units and in the emergency department (from 19% to 17%, *p* = 0.0029 and from 11% to 5%, *p* < 0.0001, respectively). The frequency of patients receiving a manipulated rectal medicine also decreased between the study years, both in inpatient units and in the emergency department (from 22% to 10%, *p* < 0.0001 and from 35% to 7% 2019, *p* < 0.0001, respectively). The results show a decrease in the manipulation of both oral and rectal medicines to paediatric patients in 2019 compared to 2009. Even though this implies a safer practice, there is still a pronounced lack of child-friendly dosage forms and suitable strengths enabling the safe administration of medicines to sick children.

## 1. Introduction

The lack of child-friendly dosage forms and strengths suitable for treating children of different ages, weights, and with different medicine-taking capabilities is a major problem in paediatric care. Consequently, manipulations of dosage forms are needed to administer prescribed doses. The definition of manipulation is the physical alteration of a medicine, often with the intention to give a proportion of the dosage form, but sometimes to facilitate intake. Examples of physical alterations are crushing or splitting a tablet, splitting or melting a suppository, opening a capsule, dissolving a tablet, taking a part of the solution, or cutting a transdermal patch. Dilution of oral or intravenous solutions is in some studies included in the definition [1,2,3].

Previous studies report frequencies of manipulation of medicines from 10% to 57% in paediatric settings [4,5], depending on the definition of manipulation and whether frequencies have been based on number of administrations or number of patients. Some studies include sub analyses of reasons for manipulation [4,5,6,7,8]. Comparisons of the manipulations with information in the Summary of Product Characteristics (SmPC) to evaluate whether the manipulation is on-label or off-label (i.e., authorised, or non-authorised) have also been performed [5,6].

The frequencies of manipulated doses, including those made to facilitate drug administration, ranged from 11% in adolescents to 64% in toddlers, with an overall average frequency of 16% [9]. These results contradict earlier reports where children older than six years have the highest frequency of manipulations [7]. The need to manipulate doses was higher in paediatric wards as compared to adult wards (50% and 17%, respectively) [10].

Paediatric and geriatric patients share some similar features regarding medicine [11]. In the elderly population, manipulations are primarily made to facilitate swallowing and not with the purpose of dose adjustments, as is one of the primary reasons in the paediatric setting. A Norwegian study reported the frequency of manipulated medicines to elderly patients, focusing on crushed or split solid dosage forms. Of the patients, 10% received at least one inappropriately altered medication and 23% received at least one drug mixed with food or beverages, which might jeopardise the effect of the medicines [12].

Several studies have focused on the dosing accuracy of split tablets, both from a paediatric perspective and for older patients, but there is only scant information regarding the clinical relevance of this behaviour [13,14,15,16]. Splitting tablets is sometimes used to save money, either in the adult population in countries where all strengths of a tablet have the same cost, or in the paediatric population where solid dosage forms are often cheaper than liquids [17,18]. There is also a risk of altered pharmacokinetics when a dosage form is manipulated [19].

When tablets are dispersed in a liquid and only a proportion is taken, the actual dose may differ markedly—3% to 99%—from the intended dose. Factors influencing actual dose include tablet characteristics, stirring technique, and extraction method [20,21]. There are almost no reports on the manipulation of solid rectal dosage forms; there is, however, one study concluding that torpedo-shaped suppositories should only be administered as intact suppositories due to difficulties in achieving the accurate target dose after splitting [22].

The European Medicines Agency (EMA) stated in 2006 that manipulation of adult dosage forms for paediatric use should be the last resort, but that it is unavoidable in many cases [23]. The manipulation of medicines to achieve a prescribed dose points to a lack of dosage forms and strengths suitable for paediatric patients. The younger the patient, the less commercial oral drugs seem to be available to meet their needs [24].

On the first of January 2007, the EU Paediatric Regulation was implemented. The main aims were to achieve more data regarding the use of medicines in children to improve the information available to prescribers and families and to encourage drug companies to develop more child-friendly dosage forms and medicine strengths [25].

### Aim

The aim of this study is to describe and compare the frequency of manipulations of solid oral and rectal medicines in 2009 and 2019 at inpatient units and an emergency department in a paediatric hospital within a Swedish university hospital.

Manipulation of medicines to paediatric patients is common practice due to the lack of child-friendly dosage forms and strengths suitable for treating children of different ages, weights, and with different medicine-taking capabilities. We hypothesise that, due to the European Paediatric Regulation, there will be more child-friendly dosage forms available in 2019 as compared to 2009 and thus the need to manipulate medicines will have decreased. We also hypothesise that patients in the emergency department receive less manipulated medicines than inpatients due to the limited number of medicines and shorter hospital stay.

## 2. Results

The number of administrations and patients in each setting for each study year are presented in Table 1. The material is divided into inpatient units and emergency department, oral and rectal administrations.

Most patients were males in all groups in both study years (55–58%) and, as patients with no documentation of sex were excluded, the remainder were females.

### 2.1. Manipulation of Oral Medicines

The frequency of inpatients with at least one manipulated solid oral medicine was 19% and 17% in 2009 and 2019, respectively (*p* = 0.0029). In the emergency department, the frequency decreased from 11% in 2009 to 5% in 2019 (*p* < 0.0001) (Table 1). The highest frequency of patients receiving an order for part of a solid oral medicine was seen in the age groups of 6–<12 years of age and 12–<18 years for both study years in both the inpatient units and the emergency department. Almost no patients younger than 2-years-old received manipulated solid oral medicines in the emergency department during the two study years. (Figure 1).

The percentages of manipulations including halves of oral dosage forms (i.e., 0.5; 1.5, etc.) are presented in Table 1. The remainder of the manipulations were for odd parts of solid dosage forms, e.g., 0.3 tablet. Of these odd parts of solid dosage forms, some were prescribed as mL, sometimes with an instruction of how the tablet should be dissolved in some liquid and a part of the solution administered (Table 1). Doses where the whole dissolved tablet or dose sachet was to be administered were not included in this percentage.

Almost all manipulations for solid oral dosage forms refer to a few ATC groups, although they differ between the two study years and ward-level. Figure 2a shows that the distribution of administrations between different ATC-groups is similar between the two study years; however, in 2009, there is an over-representation of manipulations in ATC-groups A (alimentary tract) and C (cardiovascular system), and there are more manipulations from ATC-groups H (systemic hormonal preparations) and N (nervous system) in 2019 (Figure 2a,b).

In 2009, the ATC-groups A (alimentary tract), C (cardiovascular system), H (systemic hormonal preparations), and N (nervous system) made up 85% of all manipulations for inpatients. In 2019, therapeutic groups C (cardiovascular system), H (systemic hormonal preparations), M (musculoskeletal system), and N (nervous system) represent a little more than 82% of all manipulated oral medicines administered to inpatients (Figure 2b).

Figure 3a show that the distribution of all oral administrations between different ATC-groups in the emergency department is similar between the study years, with most doses administered from ATC-groups M (musculoskeletal system) and N (nervous system).

ATC-groups M (musculoskeletal system) and N (nervous system) comprised 98.5% of all manipulated oral administrations in 2009. In 2019, ATC N (nervous system) alone represents 94.5% of all manipulated oral administrations (Figure 3b). The most frequently manipulated substance in the emergency department was paracetamol in both study years, contributing to 74% and 92% of all manipulations in 2009 and 2019, respectively.

Ten substances make up more than half of all oral solid manipulations in the inpatient setting (53% in 2009 and 54.2% in 2019) (Table 2). Sildenafil is the most frequently manipulated substance in 2009 but there are no manipulations in 2019. Paracetamol is dominating oral manipulations in both years: 9% and 11% of all manipulations 2009 and 2019, respectively.

### 2.2. Manipulation of Rectal Medicines

The frequency of inpatients with at least one manipulated solid rectal medicine was 22% and 10%, in 2009 and 2019, respectively (*p* < 0.0001). In the emergency department, the frequency decreased from 35% in 2009 to 7% in 2019 (*p* < 0.0001) (Table 1). The highest frequencies of manipulated rectal medicines were observed in patients younger than 2 years of age in both study years and in both inpatients and patients at the emergency department (Figure 4).

Inpatients in 2009 mainly received parts of solid rectal preparations from the ATC groups M (musculoskeletal system) and N (nervous system) and, in 2019, from the therapeutic groups A (alimentary tract) and N (nervous system) (Figure 5a). At the emergency department, a more pronounced increase in manipulation of ATC group A (alimentary tract) was seen between 2009 and 2019, and a similar decrease in manipulation of M (musculoskeletal system) as was seen for inpatients (Figure 5b). ATC groups M and N (musculoskeletal and nervous system) mostly contains suppositories; in ATC group A (alimentary tract), most preparations are enemas.

## 3. Materials and Methods

This registry-based, retrospective study was conducted at the Astrid Lindgren Children’s Hospital (ALB), a large Swedish university children’s hospital within Karolinska University Hospital. The hospital generally admits patients from 0 to 18 years of age and has eight departments with different specialties such as surgery, orthopaedic, oncology, intensive care, neonatology, and medicine. At the time of the study, ALB had a capacity of 250 beds, had around 2000 employees, and provided care for approximately 25% of all children in Sweden.

Data were extracted from the hospital based electronic data registry from the largest electronic prescribing record, TakeCare (KarDa). Data regarding all oral (including via feeding tube), buccal, and rectal administrations of medicines were collected for all patients 1 month–<18 years old during the years 2009 and 2019, respectively. The electronic prescribing record was introduced at Karolinska University Hospital in late 2008, and 2009 was the first complete year with data.

Patients in the intensive care units (neonatal and paediatric) mostly receive medicines intravenously and were thus excluded.

Patients with no documentation of sex were excluded (*n* = 111 (0.7%) and *n* = 93 (0.4%) in 2009 and 2019, respectively). Patients with no documentation of age but a given body weight (*n* = 0 in 2009 and *n* = 290 (1.2%) in 2019) were assumed to have an age corresponding to the weight according to the growth chart in the electronic medical record. All data were pseudonymised, i.e., patients have a unique identification number making it possible to link all administrations to a single patient.

The relevant age-groups proposed by the EMA (2001) (infants and toddlers, children, and adolescents) were used with the further division of age-groups infants and toddlers (1 month to 2 years) and children (2 to 11 years) into two groups, respectively, to better reflect differences in a child’s ability to take solid oral formulations.

All data was thoroughly scrutinised by the first author for mistakes and omissions (a few substances had an incorrect ATC-code). Some ATC-codes were changed to represent the main indication in paediatrics; for example, sildenafil was classified as C02 (antihypertensives for pulmonary hypertension), clonidine as N02 (analgesics), and naloxone given orally was classified as A06 (drugs for constipation). Concentrated electrolytes given orally were classified as A12 (mineral supplements).

In this study the definition of manipulation is the fragmenting of solid dosage forms, oral or rectal, given the prescribed dose. This includes both parts of solid dosage forms and solid dosage forms (tablets, capsules) prescribed as mL, implying that the tablet is dissolved in water. For rectal administrations, both suppositories and unit enemas were classified as solid dosage forms.

In Sweden, all these manipulations will be made by staff in the units, not by the hospital pharmacy.

Several administrations in 2019 had comments made by registered nurses that they, instead of the prescribed half suppository, e.g., had administered oral solutions or a combination of suppositories with different strengths. These administrations are included as manipulated in this study since they were prescribed as a physical alteration of a medicine.

Solid manipulated oral and rectal medicines were analysed in relation to patient age, sex, and the ATC-group of the medicine. The number of individual patients were counted in each study group (inpatients with oral administrations in 2009, inpatients with rectal administrations in 2009, patients at emergency department with oral administrations in 2009, etc.). The same patient might be included in more than one study group. When analysing data from individual patients, the age at the first occurrence of a patient in the material was recorded as the age of that individual.

Data for administrations and patients at the emergency department were analysed separately from inpatients as the level of care given and medicines prescribed at the emergency department differs from inpatient units; in the emergency setting, the patients stay for a short time, and the focus is on treating acute illness with appropriate medicines rather than administering the patients’ regular medicines.

Data were analysed for female and male patients separately; however, since there were no major differences, the results are presented in age groups only.

MS Excel (Microsoft, Redmond, WA) was used for initial descriptive statistics and GraphPad Prism version 5.04 (Graph Pad Software, San Diego, CA, USA) was used for further statistical analysis. Differences in proportions were compared by the chi-squared test. Significance was defined as *p* < 0.05. Reported *p*-values are from two-sided tests.

## 4. Discussion

Between the study years 2009 and 2019, there was a significant decrease in the frequency of patients receiving at least one manipulated oral or rectal medicine, both in the inpatient and emergency setting.

Manipulation can be authorised, i.e., supported by information in the summary of product characteristics (SmPC), or unauthorised. Unauthorised manipulation might cause dose inaccuracy and changed effects, including side effects. The clinical relevance of unauthorised manipulation depends on the formulation, type of manipulation, a narrow or broad therapeutic range drug, and the clinical indication. To manipulate medicines is very time consuming compared to just administering a whole tablet or capsule that the patient swallows. Manipulating medicines puts the nurses and caregivers at risk for environmental hazard when exposed to the substance. To study possible benefits in terms of higher patient safety with less manipulated medicines, prospective data collection regarding safety matters will be needed. This study has not had such data registered.

In this study, the frequency of manipulated solid oral dosage forms including halved tablets etc. was counted. This is not the same as on-label or authorised manipulation as not all tablets of all brands have data supporting this practice; however, splitting a tablet in halves often gives a more correct dose than splitting into quarters or smaller parts [16]. In the inpatient units, approximately a third of all manipulations were for other than a halved tablet in 2009 and approximately a fourth in 2019. In the emergency setting, on the other hand, almost all manipulations were for halved tablets.

Some of the odd manipulations include solid dosage forms prescribed as liquids in mL. In the inpatient setting 2009, 15% of all manipulations were for tablets or dose sachets dissolved and part-administered and, in 2019, the corresponding figure was 8%. In the emergency department, there were only single administrations each year, corresponding to 0.2% of all manipulations. Depending on how a tablet is dissolved and which technique is used for withdrawal of the liquid, the dose can vary many times from the expected [20]. For rectal manipulations, no frequency was calculated for halves since splitting of suppositories is not recommended.

### 4.1. EU Paediatric Regulation

The EU Paediatric Regulation has been implemented since 2007. One study from Finland investigated the effect of the Paediatric Regulation on off-label administrations (i.e., use outside of authorisation regarding indication, age and/or route) of medicines to children, but they found no difference, probably because the study was performed in 2011 when the regulation had only been effective for a few years [26]. In the present study the first study year, 2009, seems to be too close to the implementation of the regulation to demonstrate an impact at the paediatric wards. Some of the differences seen in our study could be due to the implementation of the Paediatric Regulation, but our results from 2019, 12 years after the implementation, show that there is still a lack of child-friendly medicines. Ten years after the implementation of the regulation, a compilation showed a positive and promising development of new drugs and new pharmaceutical forms, but the incentive for drug companies to perform studies in children on older off-patent drugs seemed to be low [27,28]. Even though there are more paediatric medicinal products listed in the 10-year report, the authorisation of medicines is not always the same as the marketing of medicines, thus affecting the availability in different countries, as shown in a Nordic study [29].

### 4.2. Inpatient Setting versus Emergency Department

The frequency of patients with at least one manipulated solid oral administration was lower in the emergency department compared with the inpatient units in both study years. For rectal administrations, a third of all patients received a manipulated solid dosage form 2009 in the emergency department, compared to approximately a fifth of the inpatients. In 2019, manipulated rectal administrations were lower in the emergency department than in the inpatient unit. Our hypothesis that less patients would receive a manipulated medicine in the emergency department was true for oral administrations in both study years and for rectal administrations 2019.

The number of patients with oral administrations at the emergency department was almost three times higher in 2019 than in 2009, partly due to organisational changes within the hospital between the study years. One part of the emergency department did not organisationally belong to the children’s hospital in 2009, and there was probably not a fully implemented practice to document administered medicines in the electronic prescribing record.

The number of inpatients with oral administrations are approximately the same in both study years, but there are fewer patients receiving rectal administrations in 2019. This could be caused by a shift in available medicines, but it could also be due to a change in practice, e.g., a recent recommendation to give paracetamol as an analgesic intravenously or orally instead of rectally.

### 4.3. Manipulation of Oral Solid Dosage Forms

Children aged 6 years and older received the highest frequency of manipulated solid oral medicines in our study during both study years and in both inpatient and emergency departments in accordance with results from other studies defining manipulations as primary fragmenting solid oral dosage forms [7,8]. In contrast, in studies where the definition of manipulation also includes administration of small volumes of oral liquids or dilution of oral liquids, the frequency of manipulation is usually highest in the youngest age group [5,10].

Many of the substances making up approximately 50% of all manipulated solid oral administrations in inpatients are the same in 2009 as in 2019, but there are also some interesting differences. Some of the differences in manipulated substances between the study years have a clear link to available products. Sildenafil (cardiovascular system in this study) was the most frequently (ten percent) manipulated substance in inpatients, compared to no manipulations for this substance in 2019 due to the introduction of a registered powder for oral suspension (Revatio^®^) with the label pulmonary hypertension (Table 2). In a Chinese study from 2021, the authors concluded that the lack of a child-friendly product in their country lead to frequent manipulations of sildenafil [30].

Paracetamol was the most frequently prescribed substance in both years and though only 7% and 5% of all paracetamol administrations were manipulated, it will make a large contribution to all manipulated oral solid medicines. Even though paracetamol is also a substance available in many formulations such as orodispersible tablets and an oral liquid, in both study years there were obviously still situations when parts of tablets are prescribed.

Prednisolon (systemic hormonal preparations), baclofen (musculoskeletal system), and clobazam (nervous system) are oral substances that were frequently manipulated in both study years and substances of which manipulation has increased from 2009 to 2019 (*p* < 0.0001). There were no child-friendly dosage forms with these three substances, neither in 2009 nor in 2019. Baclofen was identified already in 1999 as a drug lacking a liquid dosage form available for use in paediatric patients [31]. For dexamethasone, there was an increase in the total number of oral administrations between the study years but a decrease in manipulations (*p* = 0.0002), most likely due to better availability or knowledge of the oral solutions. Esomeprazole is the most frequently manipulated proton pump inhibitor both study years, and one of the substances where the percentage of manipulations within the substance has decreased between the study years, but since the total number of manipulations has decreased esomeprazole contributes to a higher percentage 2019 compared to 2009.

In the inpatient setting in 2009, three out of nine substances constituting 50% of all oral manipulations were from the cardiovascular therapeutic category (spironolactone, sildenafil, and hydrochlorotiazide) and one out of nine in 2019 (spironolactone). This lack of child-friendly medicines for cardiovascular conditions, leading to off-label use and manipulation of medicines, is well known [7,32,33].

### 4.4. Manipulations of Rectal Solid Dosage Forms

Children younger than 2 years of age were more frequently prescribed manipulated solid rectal medicines in both study years and in both settings, although the frequency diminished in all age groups in 2019 compared to in 2009. This probably reflects a choice of administration route as rectal administration is more common in younger children, but all available strengths are not suitable for that age group [34].

Manipulations of solid rectal administrations not only decreased between 2009 and 2019, but there was also a change in ATC-groups and substances. In Sweden, all registered suppositories are torpedo-shaped in both study years, implying that they should not be split at all. In 2009, most manipulations were parts of suppositories but in 2019 there were more parts of unit enemas; for example, ibuprofen (musculoskeletal system) suppositories were available as suppositories of 125 mg in 2009, but in 2019 the only available strength was 60 mg; hence, almost no manipulated doses were prescribed during 2019. Taking a part of an enema for constipation can be less unsafe for the patient than cutting a suppository for nervous system conditions, but it is still a messy and inconvenient procedure leading to imprecise dosing.

At ALB, there is an active Paediatric Drug Group which started writing order sets for prescribing medicines in the electronic health record in 2008. This group has been active during both study years but had an increased number of employees and greater mandate in 2019. Prefilled order sets for medicines that nurses can give in certain circumstances were written by the health care staff themselves in 2009, and they were prefilled with doses for different patient weights. Some of the rectal ones included parts of suppositories. In 2019, the Paediatric Drug Group had been responsible for these prefilled order sets for a couple of years and no split suppositories were included among the doses.

### 4.5. Strengths and Weaknesses

The strengths of this study are the large material, including administrations from two full years at a large children’s hospital, and that data regarding oral and rectal administrations are analysed separately. To our knowledge, this is the first study to separate manipulated solid oral and rectal dosage forms and to study two separate years. Most previous studies of drug manipulation are observational studies performed during short periods only. A French study looked at all oral drug dosage forms administered during one year in a children’s hospital, including more than 117,000 doses. The authors focused on administration practices but unfortunately did not have access to data regarding manipulation of dosage forms, which they also stated as a limitation of the study [35].

The limitations of this study are that frequencies of manipulated solid oral and rectal administrations have been counted from administrations registered in the electronic patient record. Therefore, the results can represent an underestimation, since manipulations made to facilitate drug intake are not included, or it can be an overestimation, since observations of what occurred were not possible. We do not know whether a manipulation took place, or if the registered nurse rounded the dose off to nearest available strength. It is also a study performed at one Swedish paediatric hospital with an active Paediatric Drug Group and ward pharmacists employed between the study years and therefore conclusions cannot automatically be generalised to other settings.

Patients with no documented sex were excluded; however, since they correspond to less than one percent, the inclusion of these data would not have affected the results in the present study.

Alteration of the pharmacokinetics is a risk when a tablet is crushed and dissolved before intake [19]. For most substances, little is known about this. Since it is not likely that there will be a full range of strengths covering the whole paediatric age, there is a need to determine the best way of administering different substances and dosage forms. In future studies, it would be of interest to compare pharmacokinetic parameters when giving a solid dose compared to a dissolved tablet and when giving a licensed oral solution compared to a manipulated tablet. It would also be interesting to study what impact ward pharmacists have on how registered nurses and doctors handle the prescription and manipulation of drugs, and how registered nurses and pharmacists reflect on doses that are not feasible to prepare from available medicines.

We do not have any data linking the administration of a manipulated dose to the clinical outcome. With this study design, it is difficult to determine the clinical relevance of a dose that is not exact due to manipulation, or to connect manipulation to any side effects occurring for the patient.

## 5. Conclusions

The frequencies of manipulation of solid oral dosage forms have decreased between 2009 and 2019, both for inpatients and patients at the emergency department. The manipulation of rectal solid dosage forms has also decreased markedly, and there are almost no parts of suppositories prescribed in 2019. In some therapeutic groups, the introduction of new drugs has diminished the need for manipulations. There is, however, still a pronounced lack of child-friendly dosage forms and suitable strengths enabling safe administration of medicines to sick children.

## Figures and Tables

**Figure 1 pharmaceuticals-16-00008-f001:**
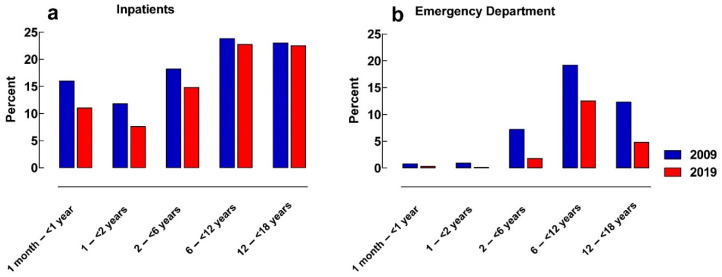
Frequency of patients with manipulated solid oral medicines in different age-groups in: (**a**) inpatients; (**b**) patients at the emergency department.

**Figure 2 pharmaceuticals-16-00008-f002:**
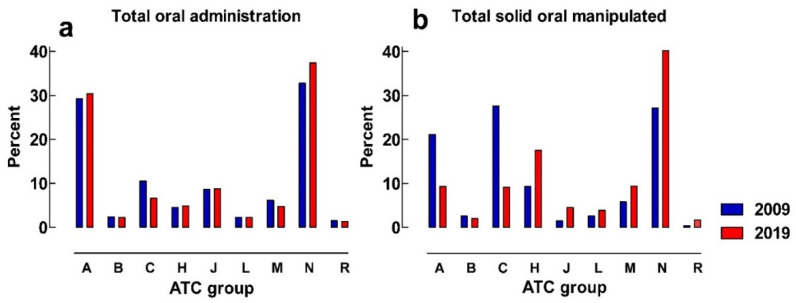
(**a**) Percentage of the distribution of total oral administrations between ATC-groups at inpatient units; (**b**) percentage of the distribution of solid oral manipulations between different ATC-groups at inpatient units. A = alimentary tract, B = blood and blood forming organs, C = cardiovascular system, H = systemic hormonal preparations, J = anti-infectives for systemic use, L = antineoplastic and immunomodulating agents, M = musculoskeletal system, N = nervous system, R = respiratory system.

**Figure 3 pharmaceuticals-16-00008-f003:**
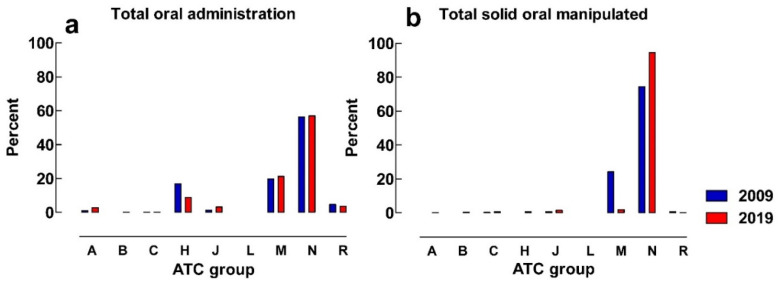
(**a**) Percentage of the distribution of total oral administrations between ATC-groups in the emergency department; (**b**) percentage of the distribution of solid oral manipulations between different ATC-groups in the emergency department. A = alimentary tract, B = blood and blood forming organs, C = cardiovascular system, H = systemic hormonal preparations, J = anti-infectives for systemic use, L = antineoplastic and immunomodulating agents, M = musculoskeletal system, N = nervous system, R = respiratory system.

**Figure 4 pharmaceuticals-16-00008-f004:**
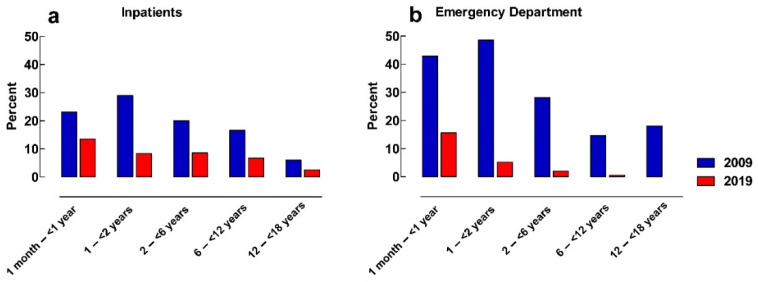
Frequency of patients with manipulated solid rectal medicines in different age-groups in: (**a**) inpatients; (**b**) patients at the emergency department.

**Figure 5 pharmaceuticals-16-00008-f005:**
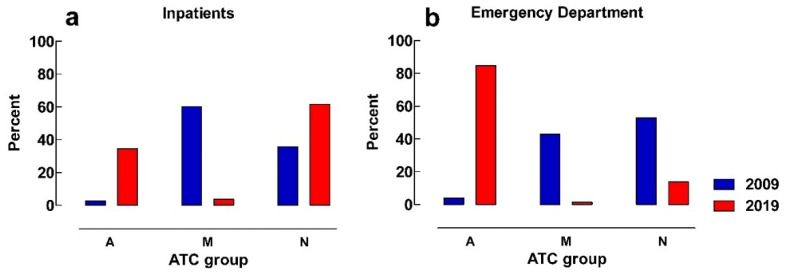
Percentage of the distribution of solid oral manipulations between different ATC-groups in: (**a**) inpatients, (**b**) emergency department; A = alimentary tract, M = musculoskeletal system, N = nervous system.

**Table 1 pharmaceuticals-16-00008-t001:** Background data and main results.

	ORAL ADMINISTRATIONS				RECTAL ADMINISTRATIONS			
	Inpatient Units		Emergency Department		Inpatient Units		Emergency Department	
	2009	2019	2009	2019	2009	2019	2009	2019
Total number of patients	4905	4718	5260	15,038	2355	1240	3883	5902
1 month–<1 year, *n* (%)	856 (17.5)	876 (18.6)	370 (7.0)	1547 (10.3)	706 (30.0)	480 (38.7)	986 (25.4)	2065 (35.0)
1–<2 years, *n* (%)	661 (13.5)	489 (10.4)	535 (10.2)	2015 (13.4)	544 (23.1)	242 (19.5)	1044 (26.9)	1548 (26.2)
2–<6 years, *n* (%)	1045 (21.3)	1097 (23.3)	1446 (27.5)	4384 (29.2)	650 (27.6)	303 (24.4)	1164 (30.0)	1624 (27.5)
6–<12 years, *n* (%)	1122 (22.9)	1259 (26.7)	1629 (31.0)	4471 (29.7)	289 (12.3)	134 (10.8)	500 (12.9)	528 (8.9)
12–<18 years, *n* (%)	1221 (24.9)	997 (21.1)	1280 (24.3)	2621 (17.4)	166 (7.0)	81 (6.5)	189 (4.9)	137 (2.3)
Male patients, %	56	56	58	55	55	57	57	55
Number of administrations	117,023	128,638	6680	24,013	12,449	5315	4639	9979
Number of administrations/patients	24	27	1.3	1.6	5.3	4.3	1.2	1.7
Patients with solid manipulated medicines, *n* (%)	953 (19)	805 (17)	581 (11)	767 (5)	509 (22)	122 (10)	1362 (35)	438 (7)
Manipulations including halves 0.5; 1.5, etc., %	64	72	97	97	NA	NA	NA	NA
Solid dosage forms prescribed as mL, %	15	8	0.2	0.2	NA	NA	NA	NA

NA = not applicable.

**Table 2 pharmaceuticals-16-00008-t002:** Substances contributing to more than 50% of all solid oral manipulated medicines in 2009 and 2019 for inpatients.

Substance	All Oral Administrations, *n*		Solid Manipulated, *n*		% Manipulated *		*p*-Value	% of All Solid Manipulations	
	2009	2019	2009	2019	2009	2019		2009	2019
Paracetamol	14,751	18,794	1025	951	7	5	<0.0001	9	11
Clonidine	9 848	12,287	568	323	6	3	<0.0001	5	4
Prednisolone	2963	2221	697	696	24	31	<0.0001	6	8
Spironolactone	1719	1073	672	230	39	21	<0.0001	6	3
Sildenafil	1645	483	1175	0	71	0	<0.0001	10	0
Baclofen	1555	1105	483	670	31	61	<0.0001	4	8
Esomeprazole	1109	2530	584	655	53	26	<0.0001	5	8
Hydrochlorotiazide	639	16	572	15	90	94	1.000	5	0.2
Clobazam	523	954	265	726	51	76	<0.0001	2	8
Dexamethasone	494	1772	137	352	27	20	0.0002	1	4

* of all oral administrations within substance.

## Data Availability

Data is contained within the article.

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
