# Peer review of "Manipulated Oral and Rectal Drugs in a Paediatric Swedish University Hospital, a Registry-Based Study Comparing Two Study-Years, Ten Years Apart"

_pharmaceuticals, 2022, doi:10.3390/ph16010008_

Round 1

Reviewer 1 Report (Previous Reviewer 3)

Dear Editors,

The authors corrected the text following the recommendations of the reviewers. I suggest accepting the manuscript for publication.

Author Response

We thank the Reviewer for this comment.

Reviewer 2 Report (Previous Reviewer 1)

I still believe that statistical analysis should only be done with one software. The use of two statistical software to perform simple statistical tests shows the lack of mastery of the authors in the science of statistics.

Author Response

We thank the reviewer for this comment. Since the two programs were not used to perform the same statistics, we have now clarified that Excel was used for initial descriptive statistics, in the last section of Materials and Methods. 

This manuscript is a resubmission of an earlier submission. The following is a list of the peer review reports and author responses from that submission.

Round 1

Reviewer 1 Report

Manuscript lines are not numbered and it is difficult to refer to the text.

The manuscript needs a major revision.

This research has no ethical code.

The writing of materials and methods is very basic and not written scientifically.

It is mentioned in the title "Ten-year outlook", but only the data of 2009 and 2019 are examined. If a 10-year period is examined, Table 1 is drawn incorrectly because it only shows data for 2009 and 2019. Also line 2 and 3.

Is the deletion of part of the information due to different electronic prescribing system a valid reason?

Why are MS Excel and GraphPad Prism software used for statistical analysis? While GraphPad Prism is a powerful software for statistical analysis.

Table 1: Why is female not reported?

No proper statistical analysis has been done on the effectiveness of different treatment methods.

Reviewer 2 Report

The paper is about manipulated medicines in a swedish paediatric hospital. In my opinion the work was well deligned and performed, with a great discussion and important references. Thus, it could be accepted in the present form.

Reviewer 3 Report

The study aimed to compare the frequencies of manipulations of solid oral and rectal medicines between 2009 and 2019 at inpatient units and an emergency department. However, it is not clear why the authors decided to compare dosage manipulations between those two years. In the Discussion section, they discuss changes in EU Paediatric Regulation, but in the Introduction section, there was no hypothesis which authors tried to analyse. They explained the main problems with the manipulation of doses in pediatric populations but didn’t explain the purpose of comparison of such data between inpatient units and emergency departments.

Also, the authors omitted patients’ data without information about gender, but they concluded that there was no difference in dosage manipulations between them. Why did the authors decide to omit such patients’ data?

On the other hand, data without information about ages were extrapolated from growth charts in the electronic medical record. Authors kindly suggest presenting percent of those data because such extrapolation can cause errors in the results.

Also, data were split into the age groups proposed by the EMA (2001) “were used with the further division of age-groups infants and toddlers (1 month to 2 years) and children (2 to 11 years) into two groups”. The authors kindly suggested explaining age categories in a more informative way. In the Result, comparisons were made between 5 age groups.

The authors kindly suggest omitting repeated data in text and tables in the results section.

The authors in Figure 2 presented only data for dosage manipulations according to ATC groups in 2009 and 2019 for hospital patients. There was no data presented on the figure for the emergency department.

In some parts of the Discussion sections, authors only repeat information from the Result sections without explaining such results (e.g. part of the Discussion related to paracetamol dosage manipulation).